# The Influence of Exposure Energy Density on Porosity and Microhardness of the SLM Additive Manufactured Elements

**DOI:** 10.3390/ma11112304

**Published:** 2018-11-16

**Authors:** Janusz Kluczyński, Lucjan Śnieżek, Krzysztof Grzelak, Janusz Mierzyński

**Affiliations:** Institute of Machine Building, Faculty of Mechanical Engineering, Military University of Technology, 00-908 Warsaw 49, Poland; lucjan.sniezek@wat.edu.pl (L.S.); krzysztof.grzelak@wat.edu.pl (K.G.); janusz.mierzynski@wat.edu.pl (J.M.)

**Keywords:** additive manufacturing, SLM technology, porosity research, microhardness research

## Abstract

Selective laser melting (SLM) is an additive manufacturing technique. It allows elements with very complex geometry to be produced using metallic powders. A geometry of manufacturing elements is based only on 3D computer-aided design (CAD) data. The metal powder is melted selectively layer by layer using an ytterbium laser. This paper contains the results of porosity and microhardness analysis made on specimens manufactured during a specially prepared process. Final analysis helped to discover connections between changing hatching distance, exposure speed and porosity. There were no significant differences in microhardness and porosity measurement results in the planes perpendicular and parallel to the machine building platform surface.

## 1. Introduction

In recent years an intensive development of additive manufacturing technology (AM) has been observed. This innovative technology is often called “3D printing”. It became one of the leading automated production technologies and it seems to be as important as subtractive manufacturing, plastic forming or casting [1]. Selective laser melting (SLM) is one of the most popular additive manufacturing techniques. It is based on selective fusion of metallic powders using an ytterbium laser, where the manufacturing process is based on a “powder bed”. During the last 10 years it has become one of the most developed AM technologies [2,3,4,5,6,7,8,9]. Regarding other additive manufacturing techniques, selective laser melting is characterized by: High-dimensional accuracy of the manufactured elements;Relatively low anisotropy of mechanical properties;A significant number of available materials;Low porosity of the manufactured elements.

The SLM process is based on low granulation powder (15–45 μm). The building job can be modified by changing different parameters which indirectly and/or directly affect the quality of the melted area. The possibilities to modify the manufacturing process in the SLM technique has created the possibility to conduct scientific research at many scientific and industry facilities [10,11,12,13,14,15,16,17,18,19]. One of the most common topics is the analysis of the process parameters which influence on the mechanical properties of manufactured elements [20,21,22,23,24,25,26,27,28,29,30]. In this paper, the influence of manufacturing process parameters on the porosity and microhardness of the additive manufactured elements was determined.

The modified parameters were:Laser power;Exposure velocity;Hatching distance.

Based on the research conducted, final conclusions were formulated and further research directions were defined.

## 2. Material

In this study, grade 316L austenitic steel (1.4404) was used. The material has been sourced by the SLM Solutions Group AG, Estlandring 4, 23560 Lübeck, Germany. Its density was 7.92 g/cm^3^. The chemical composition of the analyzed steel is shown in Table 1.

The material was manufactured using an argon atomization process. The powder particles (shown on Figure 1) have spherical or nearly spherical shapes with a particle size range between 15 μm to 45 μm. Also, satellite particles could be observed.

## 3. Experiments

Porosity and microhardness tests were carried out on specimens with the same geometry. Specimens had the form of cubes with a side length of 10 mm. These test parts were designed in such a way as to assure analysis of the distribution of mechanical properties in two different planes. The first was a plane parallel to the building platform surface, and the second one was a plane perpendicular to the platform surface.

The aforementioned planes are showed in Figure 2. As “xy” was named the plane parallel to the building platform surface, which is also normal to the direction of element growth (Z axis). The plane perpendicular to the building platform surface, which is also tangent to the direction of element growth (Z axis), is marked with “yz”.

For each sample, different sets of process parameters were used, which are summarized in Table 2. Modified parameters were components of Equation (1) which affects the additive manufacturing energy density.
(1)$$\rho_{E}\left[\frac{J}{{\mathrm{mm}}^{3}}\right]=\frac{L_{P}\left[W\right]}{e_{v}\left[\frac{\mathrm{mm}}{s}\right]\cdot h_{d}\left[\mathrm{mm}\right]\cdot l_{t}\left[\mathrm{mm}\right]}$$
where: *L_P_*—laser power [W];*e_v_*—exposure velocity [mm/s];*h_d_*—hatching distance [mm];*l_t_*—layer thickness [mm].

The modified parameters were the laser power, the exposure velocity, and the hatching distance. These specific components had been determined by the optical system and the energy source. It was caused by the possibility of analyzing the impact of modified parameters in a small range of its changes. One of the modified parameters was exposure velocity, also known as scanning speed. This determines the time of the laser exposure on each scanning line. Analysis of the influence of layer thickness on porosity and microhardness would be difficult to verify in this case for many reasons:Proper calibration of the powder reservoir (recouter);Inert gas flow speed;Clearance in the worm gear in the building platform leveling mechanism.

The manufacturing process parameters were changed within ±10% of the recommended value (item 1 in Table 2). The selected range of parameters modification was reached after consultion with specialists from the SLM Solutions company. In addition, parameters 28–30 (Table 2) differ significantly from the SLM System manufacturer’s data. The reason for testing these parameters was the good mechanical property of specimens tested and described in [31]. The specimens (Figure 3) were created during a single process. The manufacturing file for the machine was prepared using the SLM Metal Build Processor module in the Magics software (version 19.0). All specimens were manufactured using 316L austenitic steel powder.

## 4. Porosity Analysis Results and Discussion

For each specimen the porosity was analyzed in the central part of the metallographic section. All visible pores were marked in both analyzed planes. The porosity was determined by images analyzed using a scanning electron microscope (SEM) (Figure 4).

Porosity quantitative analysis were based on the microstructure images. It was carried out using a histogram check in GIMP software (version 2.0). The determination of porosity was based on the calculation of the Equation (2): (2)$$\rho [\%]=\frac{L_{p}}{L_{c}}\cdot 100\%$$
where: *L_p_*—number of pixels in the contoured pores;*L_c_*—number of pixels of the image entire area.

The porosity analysis allowed to determine the influence of used laser power, hatching distance and exposure velocity (Figure 5). The analysis includes the groups of parameters in which only one was different from the parameters tested.

Based on the conducted analysis of the laser power influence graphs, the exposure velocity and the hatching distance (Figure 5), it can be noted that the power modification has no direct effect on the porosity changes. However, the influence of the other two parameters is noticeable. During changes to the exposure velocity in the range of ±10%, the porosity changes slightly—0.02%. A significant impact on the porosity can be seen when the hatching distance changes.

To emphasize the representation of the porosity changes, depending on the exposure velocity and hatching distance, proper diagrams were plotted (Figure 6 and Figure 7).

This allows changes in the porosity in all specimens to be noticed and also the highest porosity peaks in the specimens to be noted. It was also observed that the exposure velocity and the hatching distance increase enhance the porosity growth (specimens 16 and 17 in Figure 7).

The hatching distance directly affects the porosity of the elements produced using SLM. The parameter increasing by 10% results in a nearly twice an increase in the proportion of pores. This phenomenon is caused by an increase in the distance between subsequent melt paths. This determines a reduction of the number of molten metallic powder particles. It can be observed in the parallel (xy) and perpendicular (yz) planes to the building platform surface.

A similar case can be observed when analyzing the influence of the exposure velocity, where porosity increasing with this parameter growth. It was noticed in both analyzed planes. This parameter does not affect the porosity as much as the change of the hatching distance, but with 10% change in the exposure velocity it is noticeable. Figure 6 and Figure 7 show changes in porosity related to the modification of the hatching distance and the exposure velocity for all specimens. Noteworthy are the porosity peaks for specimens 16 and 17, where both the exposure velocity and the hatching distance have been increased.

Specimens manufactured using the unusual parameters proposed in [31] did not reveal better porosity and microhardness. These settings (marked in Section 4 in Figure 7) gave very similar properties to the rest of the manufactured elements. The main feature of the settings from the study [31] was the significantly smaller hatching. Lowering this value negatively affects the process’s efficiency.

## 5. Microhardness Analysis Results and Discussion

The microhardness analysis was carried out on the same specimens used in porosity research. For each of the planes, a different configuration of the distribution of measurement points was adopted.

In the plane parallel to the building platform (“xy” in Figure 2), the influence of the linear exposure method on microhardness changes in five parallel rows was checked (Figure 8).

In the plane perpendicular to the building platform surface (“yz” in Figure 2), the effect of layers solidification (along Z axis) on microhardness changes in three parallel rows was checked (Figure 9). Similar to the porosity research, the influence of the laser power, the exposure velocity and the hatching distance (Figure 10) on the microhardness distribution in the specimens was determined. Also, in the case of microhardness analysis, the groups of parameters in which only one of the tested parameters changed were the laser power, the exposure velocity, and the hatching distance.

Proper preparation of the microhardness research allowed the lack of direct impact of parameter changes to be observed. The only observed dependence is the effect of the exposure velocity on microhardness, where microhardness decreases with the increase of the exposure velocity. However, it is insignificant and fits within the limits of measurement error. The lack of direct dependence between microhardness and one of the modified parameters was the reason for further analysis. The diagram of exposure energy density affect on microhardness was prepared. In sets where the parameters are changing in the range of ±10% from the recommended value, there is a noticeable relationship between the exposure energy density and the microhardness change (Figure 11).

Microhardness distribution on both of the measured planes helped reveal that in a range of modification of parameters by ±10% of the nominal value, microhardness slightly increases with the growth of the exposure energy density. Those changes could be connected only with exposure velocity. As recorded in Figure 11, microhardness changes are related only to the change in the exposure speed—which affects the exposure energy density. This statement is valid for the range of parameter changes within ±10% of the nominal value of the parameters only. It can be concluded that the influence of modifying manufacturing parameters on microhardness is not as important as in the case of porosity. The main reason that there are no significant changes of microhardness when parameters change is too low a range of changed laser power. In [16] a dependence between laser power, hardness (HV_0.5_) and exposure time could be noticed.

## 6. Final Conclusions

Analysis of changes in the laser power, exposure velocity and hatching distance allowed identification of the influence of these parameters on porosity and microhardness of specimens additive manufactured using the SLM technique. The research allowed the following conclusions to be drawn:There are no significant differences in microhardness and porosity measurement results in the planes perpendicular and parallel to the machine building platform surface. The main reason for the lack of visible changes of microhardness is to the low range of the changed parameters: laser power and exposure velocity;The hatching distance has a significant influence on the porosity of the manufactured elements. As the hatching distance increases, the microstructure porosity of this element increases;Exposure velocity changes affect the additive manufactured element’s porosity. Lowering the exposure velocity cause the porosity to decrease;The relationship between exposure energy density changes and microhardness was identified. In the range of ±10% of the nominal value of the parameters, an increase of microhardness with an increase of the exposure energy density was observed. The microhardness increase is connected with the combined effect of grain refinement strengthening (Hall–Petch relation) and grain boundary strengthening [22];Conducted analyses of porosity and microhardness allowed for the selection of 5 groups of parameters which will be used to produce specimens for further research.

## Figures and Tables

**Figure 1 materials-11-02304-f001:**
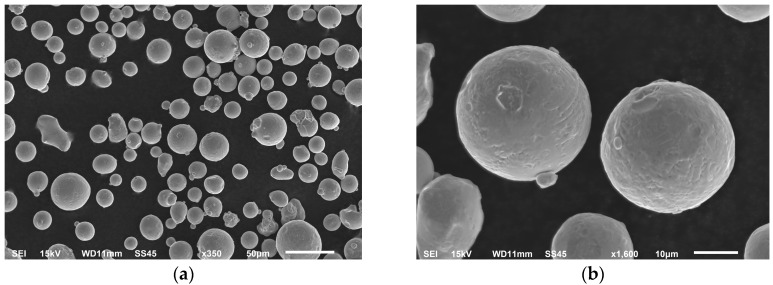
316L powder scanning electron microscope (SEM) micrographs with (**a**) 50 µm scale and (**b**) 10 µm scale.

**Figure 2 materials-11-02304-f002:**
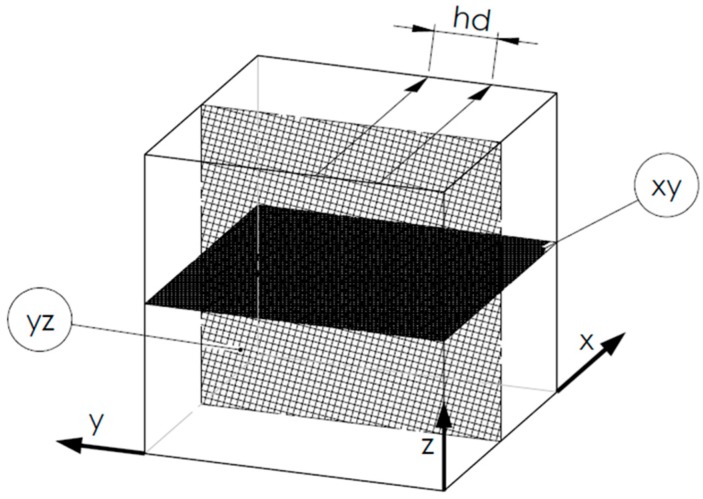
3D model of a cubic sample, where: x—plane parallel to the building platform surface, y—plane perpendicular to the building platform surface, hd (hatching distance)—distance between the exposure vectors, Z—direction of growth (element building).

**Figure 3 materials-11-02304-f003:**
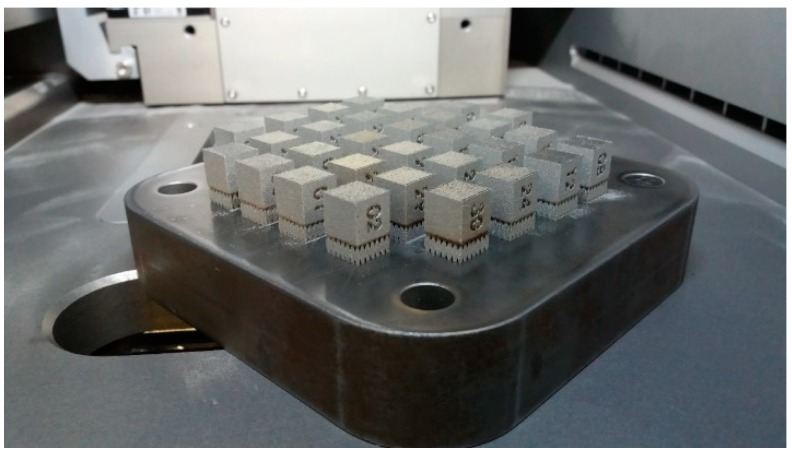
SLM 125HLs’ building platform with manufactured specimens.

**Figure 4 materials-11-02304-f004:**
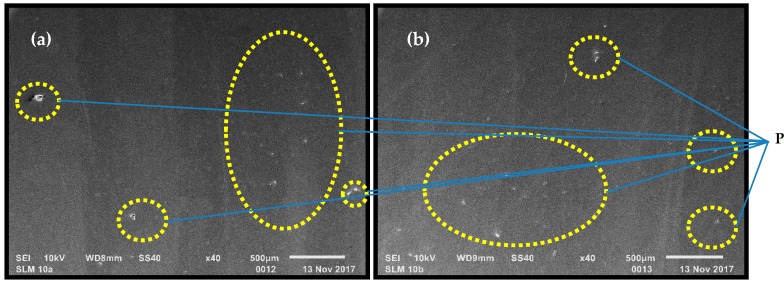
Image of visible pores in the plane parallel (**a**) and perpendicular (**b**) to building platform surface (areas of pores marked with the letter “P”).

**Figure 5 materials-11-02304-f005:**
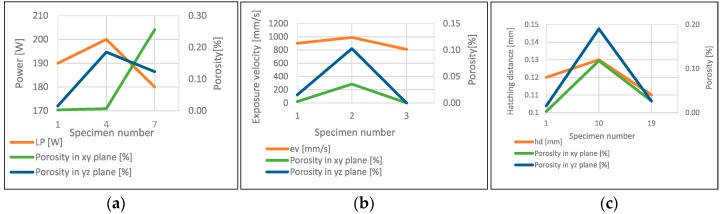
The influence of power (**a**), exposure velocity (**b**) and hatching distance (**c**) on porosity in the parallel (xy) and perpendicular plane (yz) to the building platform surface.

**Figure 6 materials-11-02304-f006:**
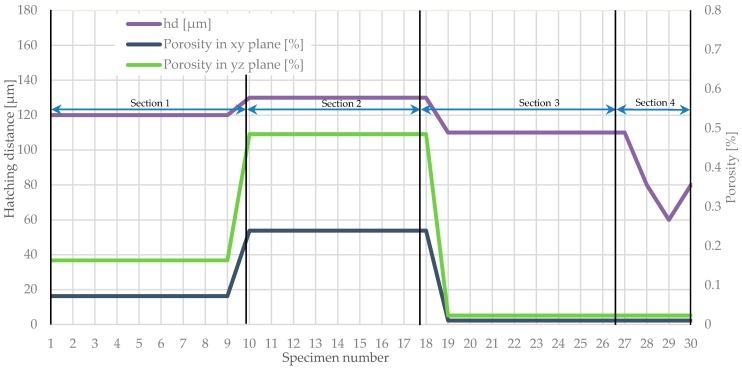
Variations of the average porosity in four ranges referenced to modified hatching distance in particular parameters group.

**Figure 7 materials-11-02304-f007:**
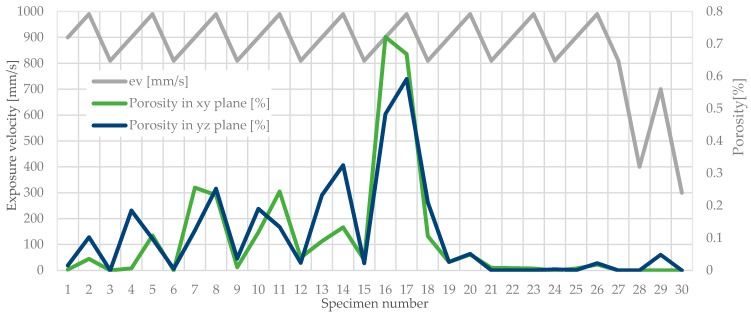
Porosity changes related to modified exposure velocity in particular parameters group.

**Figure 8 materials-11-02304-f008:**
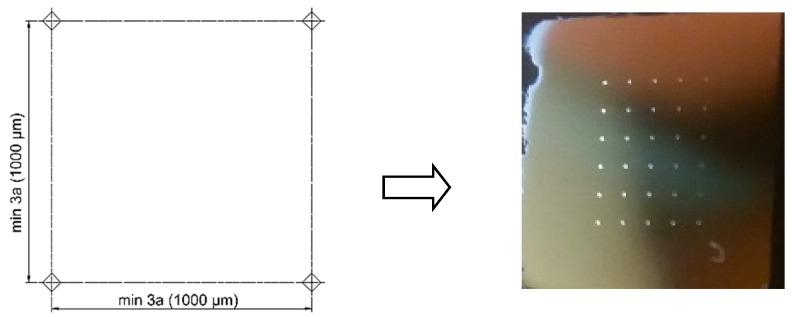
Distribution of the measurement points on a plane parallel to the building platform surface.

**Figure 9 materials-11-02304-f009:**
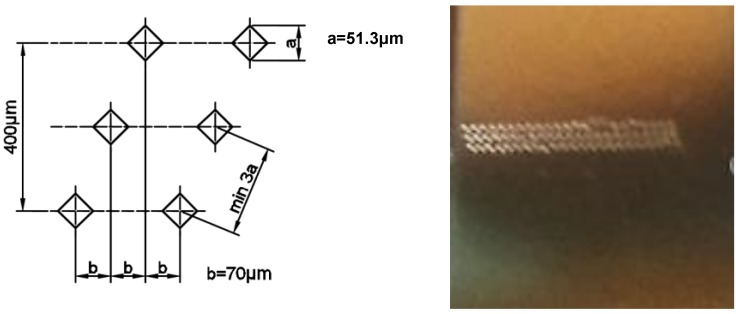
Distribution of the measurement points on a plane perpendicular to the building platform surface.

**Figure 10 materials-11-02304-f010:**
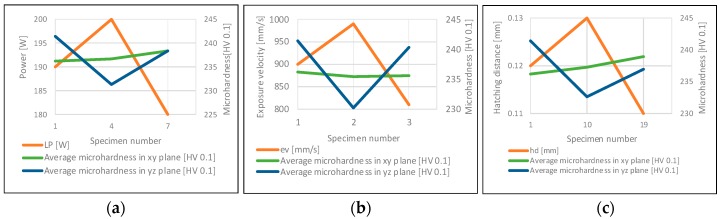
The influence of power (**a**), exposure velocity (**b**) and hatching distance (**c**) on microhardness in the parallel (xy) and perpendicular plane (yz) to the building platform surface.

**Figure 11 materials-11-02304-f011:**
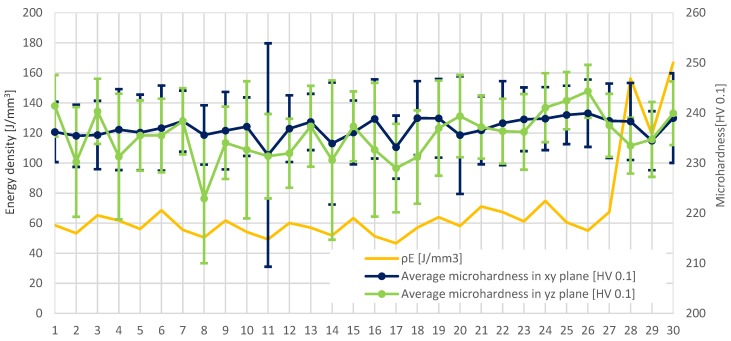
The influence of exposure energy density on microhardness in particular groups of parameters.

**Table 1 materials-11-02304-t001:** Chemical composition of 316L steel.

C	Mn	Si	P	S	N	Cr	Mo	Ni
wt.%								
max. 0.03	max. 2.00	max. 0.75	max. 0.04	max. 0.03	max. 0.10	16.00–18.00	2.00–3.00	10.00–14.00

**Table 2 materials-11-02304-t002:** Sets of analyzed production parameters.

Parameters Set	*L_P_* [W]	*e_v_* [mm/s]	*h_d_* [mm]	*ρ**_E_* [J/mm^3^]
1	190	900	0.12	58.64
2	190	990	0.12	53.31
3	190	810	0.12	65.16
4	200	900	0.12	61.73
5	200	990	0.12	56.12
6	200	810	0.12	68.59
7	180	900	0.12	55.56
8	180	990	0.12	50.51
9	180	810	0.12	61.73
10	190	900	0.13	54.13
11	190	990	0.13	49.21
12	190	810	0.13	60.15
13	200	900	0.13	56.98
14	200	990	0.13	51.80
15	200	810	0.13	63.31
16	180	900	0.13	51.28
17	180	990	0.13	46.62
18	180	810	0.13	56.98
19	190	900	0.11	63.97
20	190	990	0.11	58.16
21	190	810	0.11	71.08
22	200	900	0.11	67.34
23	200	990	0.11	61.22
24	200	810	0.11	74.82
25	180	900	0.11	60.61
26	180	990	0.11	55.10
27	180	810	0.11	67.34
28	150	400	0.08	156.25
29	150	700	0.06	119.05
30	120	300	0.08	166.67

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
