# Peer review of "The Influence of Exposure Energy Density on Porosity and Microhardness of the SLM Additive Manufactured Elements"

_materials, 2018, doi:10.3390/ma11112304_

Round 1

Reviewer 1 Report

In this manuscript, the authors studied the influence of energy density on porosity and microhardness of the SLM manufactured elements. The paper in general is interesting but require further modifications:

1.       First, you need to specify what material are you testing? This need to be specified in the title of the paper to avoid generality.

2.       Please define the term "exposure velocity" as it seem not common in the industry.

3.       Why there is no significant variation in hardness?

4.       The explanation for Figure 8 is totally not clear.

5.       There are a great deal of recently published papers concerning SLM processing of 316L and it composites. The authors need to go through the following example papers and cite them and relate their findings with previously published work:

- Electrical energy consumption and mechanical properties of selective-laser-melting-produced 316L stainless steel samples using various processing parameters

- Densification behavior, microstructural evolution, and mechanical properties of TiC/316L stainless steel nanocomposites fabricated by selective laser melting

- On the influence of laser defocusing in Selective Laser Melting of 316L

- Thermal behavior of the molten pool, microstructural evolution, and tribological performance during selective laser melting of TiC/316L stainless steel nanocomposites: Experimental and simulation methods

- Effect of selective laser melting (SLM) process parameters on microstructure and mechanical properties of 316L austenitic stainless steel

- In-situ formation of novel TiC-particle-reinforced 316L stainless steel bulk-form composites by selective laser melting

Author Response

Dear Reviewer,

We are very grateful for your review, value opinion and very useful correction tips.

In connection to your notes we have made the following corrections:

"First, you need to specify what material are you testing? This need to be specified in the title of the paper to avoid generality."
We have added the new chapter where we have made a powder micographs using our SEM and also we have described the material. 

"Please define the term "exposure velocity" as it seems not common in the industry"
We have clarify the exposure velocity term (which is similar to scanning speed). 

"Why there is no significant variation in hardness?"
We have reffered to some works where there were similar issue investigated.

"The explanation for Figure 8 is totally not clear."
We have changed the desription of the mentioned figure and add the reference to other figure where the perpendicular plane was clarify

"There are a great deal of recently published papers concerning SLM processing of 316L and it composites. The authors need to go through the following example papers and cite them and relate their findings with previously published work"
Our paper was being prepared for several months and many of mentioned positions showed in last few months. We are very grateful for those papers. We have related our findings with those authores and also made proper citations. 

We hope that our amendments have met your expectations. 

Best Regards,

Authors

Reviewer 2 Report

Authors should be consistent with referencing Figures: or Figure (line 113) or Fig. (line 143) or fig.7 (line 144).

In the Final Conclusions section 3rd bullet point might be needs a correction.

When authors saying +/- 10% (for example, line 178, line 182) some comments would be useful.

Author Response

Dear Reviewer,

We are very grateful for your review, value opinion and very useful correction tips.

In connection to your notes we have made the following corrections:

"Authors should be consistent with referencing Figures: or Figure (line 113) or Fig. (line 143) or fig.7 (line 144)"

We have corrected mentioned issue.

"In the Final Conclusions section 3rd bullet point might be needs a correction."

We made a proper corrections.

When authors saying +/- 10% (for example, line 178, line 182) some comments would be useful.

"We were add some comments to +/-10% parameters changing range description.

We hope that our amendments have met your expectations. 

Best Regards,

Authors

Reviewer 3 Report

Dear authors,

The paper is interesting for the process planning of selective laser melting operations in the particular machine used and with the particular conditions tested. However it lacks fundamental research in the topic and in the process behaviour for increasing mechanical performance. There are several concerns that prevent my recommendation to the publication. The introductory section is very basic and not suitable for specialized readers. Nowadays the SLM process is under the domain of the industry and the academic community and its instroduction is not a novelty anymore. Furthermore the study of porosity and microhardnes in metallic additively manufactured alloys has been comprehensively conducted in the recent years. The information provided after the experimentation is not conclusive and is only an output of the combination of well known parameters like hatching distance, exposure velocity and laser thickness. Additionally, the paper does not include any description of the shape, size and constitution of the powder, that is generally used for the explanation of undesirable effects like the excess of porosity. Other problems detected in the paper are the low quality of the figures and the logical arguments used thoroughly the paper.

Author Response

Dear Reviewer, 

We are very grateful for your review and value opinion.

In connection to your objections we would like to assure you we have made proper research connected with increasing mechanical properties. What is more, we have arrange a consultations with specialist form R&D Department form the SLM Solutions company - one of the leading SLM systems producers. They have rightly mentioned that there is lack of the reaserch where there is a small range of parameters modifications. In our topic research we have not find papers with that kind of parameters modifications.

With refernece to your opinion about the introductory we would like to point that we are fully conscious that it is not a novelty. As you can read there - we have mentioned that SLM techniqe became one of the most developed technique by the last 10 years. All we wanted to achive is to connect the introductory with the main topic. Very detailed descriptions are suitable for the review papers.

As we mentioned at the beggining we have made proper reasearch and discovered that there is lack of study where the main parameters were change in a small range (+/-10% of all combinations). As you surely checked we have also attached some parameters form other research papers.

We have added additional conclusions connected or related with other, similar research papers. Also we have add a chapter dedicated to the powder description.

The last thing was a figures correction (there was an issue with the one form microhardness testing method description) and increasing the substantive content of the argumentation. 

We hope that our amendments have met your expectations. 

Best Regards,

Authors

Round 2

Reviewer 1 Report

Well-done.

Reviewer 3 Report

Dear authors,

I appreciate your comments and your efforts to have your paper published. However, it still doesn't have a solid argumentation regarding the "lack of parameters". It is not recommended (I would say is not allowed) to include opinions from machinery vendors in academic research. Please provide a better support for your assumption that there is a lack of parameters regarding SLM calibration. There are several works in the field of process parameters. Please check "Ramirez-Cedillo, Erick, et al. "Process planning guidelines in selective laser melting for the manufacturing of stainless steel parts." Procedia Manufacturing 26 (2018): 973-982."